# Hepatoprotective Effects of Standardized Extracts from an Ancient Italian Apple Variety (Mela Rosa dei Monti Sibillini) against Carbon Tetrachloride (CCl_4_)-Induced Hepatotoxicity in Rats

**DOI:** 10.3390/molecules25081816

**Published:** 2020-04-15

**Authors:** Hasan Yousefi-Manesh, Ahmad Reza Dehpour, Sedighe Ansari-Nasab, Sara Hemmati, Mohammad Amin Sadeghi, Reza Hashemi Shahraki, Samira Shirooie, Seyed Mohammad Nabavi, Joice G. Nkuimi Wandjou, Stefania Sut, Giovanni Caprioli, Stefano Dall’Acqua, Filippo Maggi

**Affiliations:** 1Department of Pharmacology, School of Medicine, Tehran University of Medical Sciences, Tehran 13145-784, Iran; Hasanyousefimanesh@gmail.com (H.Y.-M.); dehpour@yahoo.com (A.R.D.); Sedighe.ansari.nasab.s.a.n@gmail.com (S.A.-N.); Sara_hem75@gmail.com (S.H.); Masadeghi6@gmail.com (M.A.S.); 2Experimental medicine research center, Tehran University of Medical Sciences, Tehran 13145-784, Iran; 3Department of Medical Physics and Biomedical Engineering, School of Medicine, Tehran University of Medical Sciences, Tehran 13145-784, Iran; rezahashemi63@gmail.com; 4Preclinical Core Facility, Tehran University of Medical Sciences, Tehran 13145-784, Iran; 5Pharmaceutical Sciences Research Center, Health Institute, Kermanshah University of Medical Sciences, Kermanshah 6715847141, Iran; 6Applied Biotechnology Research Center, Baqiyatallah University of Medical Sciences, Tehran 14359-16471, Iran; nabavi208@gmail.com; 7School of Pharmacy, University of Camerino, 62032 Camerino, Italy; joice.nkuimiwandjou@unicam.it (J.G.N.W.); giovanni.caprioli@unicam.it (G.C.); 8Department of Agronomy, Food, Natural Resources, Animals and Environment, University of Padova, 35020 Legnaro, Italy; stefania_sut@hotmail.it; 9Department of Pharmaceutical and Pharmacological Sciences, University of Padova, 35121 Padova, Italy; stefano.dallacqua@unipd.it

**Keywords:** CCl4, hepatotoxicity, apple extract, LC-MS, antioxidant, anti-inflammatory

## Abstract

The aim of this research was to examine the effect of the hydroalcoholic extracts from the peel (APE) and pulp (APP) of a traditional apple cultivar from central Italy (Mela Rosa dei Monti Sibillini) on CCl_4_-induced hepatotoxicity in rats. Phytoconstituents were determined by liquid chromatography–mass spectrometry (LC-MS) analysis showing an abundance of proanthocyanidins and flavonol derivatives together with the presence of annurcoic acid in APE. Wistar rats received APE/APP (30 mg/kg oral administration) for three days before CCl_4_ injection (2 mL/kg intraperitoneal once on the third day). Treatment with both APE and APP prior to CCl_4_ injection significantly decreased the serum levels of aspartate aminotransferase (AST), alkaline phosphatase (ALP) and alanine aminotransferase (ALT) compared to the CCl_4_ group. Besides, pretreatment with APE reversed the CCl_4_ effects on superoxide dismutase (SOD), myeloperoxidase (MPO), tumor necrosis factor-α (TNF-α) and interleukin-1beta (IL-1β) levels in liver tissue in rats and reduced tissue damage as shown in hematoxylin and eosin staining. These results showed that this ancient Italian apple is worthy of use in nutraceuticals and dietary supplements to prevent and/or protect against liver disorders.

## 1. Introduction

The Mela Rosa dei Monti Sibillini is a small ancient apple, cultivated in the pre-Apennine zone of the Sibillini Mountains in central Italy, at an altitude between 400 and 900 m a.s.l. It is considered a typical fruit of the Marche region in central Italy [1]. Currently, local authorities are encouraging various actions devoted to the increase of orchards throughout the territory of the Sibillini Mountains in order to revive the regional economy and support industrial applications in cosmetics, nutraceuticals and dietary supplements.

Ancient apples can be a valuable source of phytoconstituents, and our research group has already explored the nutraceutical potential of some Italian cultivars of north and central Italy [2]. 

Apple (*Malus* spp.), one of the most widely obtainable fruits, contains phenolic compounds belonging to the following classes: flavan-3-ols/procyanidins (catechin, epicatechin and procyanidins A, B), flavonols (quercetin derivatives), dihydrochalcones (phloretin and its glycosylated form phloridzin) and hydroxycinnamic acids (chlorogenic acid) [1,3]. Furthermore, it is known as a good source of triterpene acids, especially in the peel, such as ursolic, annurcoic and oleanolic acids [1,3,4].

Literature data from clinical in vitro and in vivo experiments and overall research suggest that apple consumption may be beneficial for the reduction of the risk of chronic diseases due to its antioxidant, anti-inflammatory, antiproliferative, and cell signaling effects [4]. Thus, the exposure to apples’ constituents derived from dietary consumption has been associated with beneficial effects on the risk, markers, and etiology of several degenerative diseases [5].

Apple peel extract (APE) and apple pulp extract (APP) have shown beneficial effects in acetic-acid-induced colitis, decreasing serum levels of triglycerides (TG), LDL cholesterol (LDL-C) and VLDL cholesterol (VLDL-C) in streptozotocin (STZ)-induced diabetic rats [4,6]. In addition, they displayed anti-inflammatory effects through suppression of NF-kB activity, inhibition of inflammatory cytokines expression and improvement of antioxidant enzymes [4,7].

Extracts enriched in phytoconstituents may also exert significant effects in some disease models and we previously observed significant protective effects of extracts from the Mela Rosa dei Monti Sibillini against renal ischemia/reperfusion injury in rats [8].

Therefore, in the proceeding of our studies related to health-promoting effects of extracts from the Mela Rosa dei Monti Sibillini we decided to evaluate their possible role on liver function protection.

The liver is the major organ devoted to the metabolism of drugs and toxicants [9]. Several factors, such as viral infections, alcohol abuse, toxic substances and drugs, have potential harmful effects on liver. Liver injury is characterized by cell degeneration, necrosis and apoptosis. Liver fibrosis and cirrhosis is the main cause of the incidence of liver cirrhosis and hepatocellular carcinoma [10]. Numerous halogenated chemicals like carbon tetrachloride (CCl_4_) have potential hepatotoxic effects due to cellular damage via oxidative stress [11]. CCl_4_ is converted to reactive radicals such as CCl_3_* and CCl_3_OO* by cytochrome P450 in the liver and these free radicals react with membrane lipids, proteins, DNA and RNA, leading to lipid peroxidation, cellular damage and apoptosis [12]. Several antioxidant enzymes including superoxide dismutase (SOD), catalase (CAT), phospholipid hydroperoxide glutathione peroxidase (GSH-Px) play a protective role by inactivating free radicals and preventing the cell damage [13]. Myeloperoxidase (MPO) is a key enzyme increasing its activity during hepatic damage due to ongoing inflammation and oxidative stress [14], thus its reduction is of pivotal importance for the treatment of liver injury. The inhibition of free radical formation and the activities of antioxidant enzymes have a significant role for the survival and proper functioning of cells. CCl_4_ intoxication is a useful model to evaluate the hepatoprotective effects of natural products obtained from medicinal plants and foodstuffs.

Extracts from medicinal and aromatic plants as well as fruits and vegetables are ready-to-use sources of antioxidant agents so that they may be proposed as an effective prevention treatment against several diseases [15]. It has been reported that polyphenolic compounds from fruits and vegetables present protective effects against tissue injury caused by free radicals [16]. 

Several studies on CCl_4_-induced hepatotoxicity in animals have highlighted promising effects for various phytochemicals [17]. For example, proanthocyanidins, natural compounds made up of polymers of flavan-3-ols occurring in grape seed extracts, have shown a hepatoprotective role in CCl_4_-induced hepatotoxicity in rats through reduction of lipid accumulation and DNA damage, and increase of the level of antioxidant enzymes [12]. As well, a pretreatment with phloretin, the main apple dihydrochalcone, significantly decreased the CCl_4_-induced inhibition of serum alanine aminotransferase (ALT), aspartate aminotransferase (AST) and lactic dehydrogenase (LDH) activities [16]. Owing to its potent antioxidant and free radical scavenging activities, quercetin ameliorated CCl_4_-induced oxidative stress and reduced levels of reactive oxygen species (ROS) and CYP2E1 expression in mice liver [18]. Similarly, a polyphenolic-rich extract from *Micromeria croatica* (Pers.) Schott showed a protective effect against CCl_4_-induced hepatotoxicity in mice by boosting SOD activity and reducing 4-hydroxynonenal (4-HNE) formation in the liver [19]. 

Thus, on the basis of the reported effects on liver of several phytoconstituents that have been reported for the Mela Rosa dei Monti Sibillini, we decided to investigate the protective effects of APE and APP against CCl_4_-induced hepatotoxicity in rats. For this purpose, the serum levels of ALT, AST and alkaline phosphatase (ALP) were measured and the levels of SOD, MPO, tumor necrosis factor (TNF-α) and interleukin-1beta (IL-1β) determined in the tissue homogenates. During CCl4 intoxication, serum creatinine and urea levels are often increased by impairment of glomerular filtration rate due to the delayed CCl4 elimination. Therefore, urea and creatinine levels were also measured in order to assess nephroprotective effects of our treatments [20]. Finally, the tissue injury was assessed by hematoxylin and eosin staining. APE and APP chemical profiles were studied by liquid chromatography coupled with mass spectrometry (LC-MS*^n^*) analysis.

The findings of our work shed light on the exploitation of this traditional apple variety in the central Italy economy as a source of nutraceuticals to be used in the prevention and treatment of liver disorders.

## 2. Results

### 2.1. Chemical Composition of APE and APP

Detailed chemical analysis of APE and APP was the subject of a previous work [1] and revealed a complex pattern of phytoconstituents comprising phloretin glycosides, hydroxycinnamic acid derivatives, quercetin derivatives and procyanidins of the B group as the most abundant phytochemicals. Furthermore, annurcoic acid was detected as one of the main triterpene acids in APE. To assess a possible role of these constituents in this work we obtained a fingerprint of the extract and we used the overall content of phenolics and triterpene acids measured in the samples used for the in vivo test. Chromatographic fingerprints of the extracts with the chemical structures of the main constituents are depicted in Figure 1 while quantitative values are reported in the Figure 2.

The overall results showed that administration of extracts to animals corresponded to a dose of 4.21 and 6.52 mg/g of APP and APE, respectively. It is worth mentioning that only APE contained annurcoic acid (0.23 mg/g).

### 2.2. Effects of APP and APE on Serum Biochemical Parameters

Table 1 shows the effects of APP and APE administration on CCl_4_-induced hepatotoxicity. The serum levels of AST (^###^
*p* < 0.001), ALT (^##^
*p* < 0.01) and ALP (^#^
*p* < 0.05) in the CCl_4_ group were significantly raised compared to the control group. The serum levels of AST (** *p* < 0.01), ALT (** *p* < 0.01) and ALP (* *p* < 0.05) were significantly lower in both the APP and APE pretreatment groups compared to the CCl_4_ group. However, other biochemical parameters for instance, urea and creatinine revealed no significant differences compared to the control group.

### 2.3. Effects of APP and APE on Liver Enzyme Assessments

Figure 3a,b shows the effects of APP and APE on liver enzymes (SOD and MPO). SOD activity was reduced significantly after injection of CCl_4_ in the CCl_4_ group compared to the control group (*** *p* < 0.001). Pretreatment with APE significantly improved the extent of SOD activity compared to the CCl_4_ group (** *p* < 0.01). Pretreatment with APP before CCl_4_ intoxication had no significant effects on SOD activity compared to the CCl_4_ group (*p* > 0.05) (Figure 3a). The levels of MPO as a pro-oxidative and pro-inflammatory marker in the CCl_4_ group was markedly elevated compared to the control group (*** *p* < 0.001). However, pretreatment with APE significantly decreased the MPO levels compared to the CCl_4_ group (* *p* < 0.05), while the pretreatment with APP had no significant effects (*p* > 0.05) (Figure 3b). The levels of pro-inflammatory cytokines of liver such as TNF-α (*** *p* < 0.001) and IL-1β (*** *p* < 0.001) were significantly increased after CCl_4_ intoxication compared to the control group (Figure 4a,b). Pretreatment with APE significantly decreased TNF-α (* *p* < 0.05) and IL-1β (*** *p* < 0.001) levels compared to the CCl_4_ group. On the other hand, pretreatment with APP before CCl_4_ administration had no significant effects on these cytokines (*p* > 0.05).

### 2.4. Histopathological Examination of the Liver

Histopathological micrographs of rat liver are presented in Figure 5. Control liver showed a normal histologic structure (**A**). In the CCl_4_-treated group, disruption of cellular and lobular structures, necrosis, congestion and inflammation were observed around the central vein (**B**). Pretreatment with APP did not show observable improvement in CCl_4_-induced pathologic features (**C**). On the other hand, the liver from APE-pretreated rats showed only mild damage suggesting hepatoprotective effects of this extract (**D**).

## 3. Discussion

It is well documented that CCl_4_ intoxication can induce liver lesion, cirrhosis and hepatocarcinoma [21,22], thus being a well-established model of liver damage. In the present study, CCl_4_ administration increased the serum levels of ALT, AST, ALP. Yousefi-Manesh et al. have reported that APE prevented kidney injury induced by ischemia and reduced oxidative stress [8]. In the present study, the pretreatment of rats with both APE and APP obtained from the same Italian apple variety (Mela Rosa dei Monti Sibillini) significantly reduced the serum levels of ALT, AST and ALP compared to the CCl_4_ group suggesting a protective effect. Furthermore, APE pretreatment gave beneficial effects on TNF-α and IL-1β levels reducing the production of these pro-inflammatory cytokines in the liver compared to the CCl_4_ group.

It has been reported that CCl_4_ intoxication increases MPO as an index of liver neutrophil infiltration and decreases SOD, which plays the main role in the antioxidant defense [23,24]. APE pretreatment was able to normalize the MPO and SOD levels whereas APP did not give significant changes in these parameters compared to the CCl_4_ group. Ohta et al. demonstrated that the liver MPO activity increased 6 h after CCl_4_ injection followed by a further increase at 24 h [23]. On the other hand, the liver SOD activity decreased 6 h after CCl_4_ injection followed by a further decrease at 24 h [23]. CCl_4_ is converted into free radicals such as CCl_3_− and Cl_3_COO− by the hepatic cytochrome P450, that contributes to elevate the serum levels of ALT, AST, ALP and MPO and to reduce the activity of antioxidant enzymes such as SOD [25]. Our data revealed that pretreatment with APE could normalize the levels of these parameters. Some reports have shown that CCl_4_ administration led to collagen deposition, destruction of cellular boundaries, central vein congestion and inflammation of liver tissue [11,25,26]. In this study, the histopathological examination of the liver indicated that CCl_4_ intoxication caused disruption of cellular and lobular structures, inflammation and necrosis. Our results showed that pretreatment with APE could decrease CCl_4_ toxicity whereas APP pretreatment had no beneficial effects. This different effects of APE and APP treatment, as observed in histopathological examination and pro-inflammatory cytokines levels, may be related to their different chemical compositions. For instance, APE is richer than APP in flavan-3-ols/procyanidins, flavonols, dihydrochalcones, hydroxycinnamic acids, dihydrochalcones and triterpenes (average contents 8.04 vs. 3.93 mg/g, respectively), with bioactive compounds such as quercetin derivatives, procyanidins, phloridzin and annurcoic acid as the most important ones [1]. Furthermore, only APE contained the triterpene annurcoic acid, suggesting a role for this compound in the observed activity. Flavonol glycosides have shown protective effects against CCl_4_ hepatotoxicity in rats by improving antioxidant parameters such as glutathione S-transferases (GSH), SOD and catalase (CAT) and decreasing the serum levels of ALT, AST and ALP [27]. Dihydrochalcones have shown hepatoprotective effects by reducing the serum levels of ALT induced by CCl_4_ intoxication and inhibiting cyclooxygenase (COX-2) and inducible nitric oxide synthase (iNOS) expression. They also downregulated the expressions of nuclear factor-κB (NF-κB), IL-6, caspase 3/8 induced by CCl_4_ [28]. Phloridzin, one of the most abundant dihydrochalcones in APE, was shown to be able to revert the methotrexate (MTX)-induced hepatotoxicity in rats by reducing oxidative stress, inflammation and apoptosis in hepatic tissues [29]. Annurcoic acid is a derivative of ursolic acid, which was found to protect against CCl_4_-induced inflammation owing to its antioxidant capacity and modulation of mitogen-activated protein kinase (MAPK) and NF-κB pathways [30].

In a research on zymosan-induced paw edema in rat, treatments with apple peel extracts resulted in anti-inflammatory effects and reduction of paw edema and production of inflammatory cytokines such as TNF-α and IL-1β [31]. Apple peel polyphenols have preventing effects against both iron ascorbate (Fe/Asc) and lipopolysaccharide (LPS)-induced oxidative stress and inflammation through downregulation of TNF-α, IL-6 and prostaglandin E2 (PGE2), and reduction of lipid peroxidation. They also increased the induction of nuclear factor erythroid 2-related factor 2 (Nrf2) and peroxisome proliferator-activated receptor-γ coactivator (PGC-1α) which elevate the expression of antioxidant proteins and protect macromolecules against reactive oxygen species (ROS) damage [32]. In vitro studies on several cell lines (DLD-1, T84, MonoMac6, Jurkat) have demonstrated that apple polyphenolic extracts had anti-inflammatory effects through NF-κB suppression and downregulation of cyclooxygenase (COX-2), chemokines CXCL9 and CXCL10, and proinflammatory cytokines [33].

## 4. Materials and Methods

### 4.1. Apple Sampling

Apple sampling was done as formerly reported by Yousefi-Manesh et al. (2019). Apple trees were cultivated in the orchards located in Montefalcone Appennino (N 42°59′17″; E 13°27′32″), in the area of Sibillini Mountains, Marche region, central Italy, at 750 m a.s.l. The harvest was done at apple ripening, in November 2017. Apple storage was at ambient temperature until extraction.

### 4.2. Preparation of Hydroalcoholic Extracts

The hydroalcoholic extracts were prepared as formerly reported by Yousefi-Manesh et al. (2019). Briefly, after the separation and drying of peel and pulp at 40 °C for 18 h with a Biosec De Luxe B12 dryer (Albrigi Luigi, Verona, Italy), the dehydrated materials were powdered to 2-mm size particles using an IKA-WERK MFC DCFH 48 (Staufen, Germany). The extraction was made using a methanol/water solution (1:1 *v*/*v*) with the dehydrated material in an ultrasound bath at 45 °C for 60 min. After collection, a part of the liquid extract was used for HPLC-DAD-MS analysis while the remaining was concentrated by a rotavapor at 40 °C under vacuum to obtain the crude extracts (APE and APP) ready for animal studies.

### 4.3. HPLC-DAD-MS^n^ Analysis

HPLC-DAD-MS*^n^* was used to analyze the phenolic derivatives of the extracts. The analytical methods are the ones used in our previous works [1,8].

Composition of phenolic derivatives was obtained by HPLC-DAD-MS^n^ using an Agilent 1260 chromatograph (Santa Clara, CA, USA) with diode array (DAD) and Varian MS-500 ion trap mass spectrometer. Agilent Eclipse XDB C-18 (3.0 × 150 mm, 3.5 µm) was used as stationary phase while mobile phases were acetonitrile (A) and water with 0.1% formic acid (B). The flow rate was 500 μL/min. The gradient of elution started at 95:5 A:B then went to 85:15 A:B at 15 min, 15:85 A:B at 35 min, finally 0:100 A:B at 48 min. A ‘T’ connector divided the eluate in equal amounts to DAD and MS. The DAD detector was used to quantify phenolic compounds, and rutin, chlorogenic acid, phloridzin and catechin (Sigma-Aldrich, Milan, Italy) were used as reference compounds. UV-Vis spectra were obtained in the range of 200–650 nm. The sample injection volume was 10 μL. MS spectra were recorded in negative ion mode in 50–2000 Da range, using ESI ion source. Fragmentation of the main ionic species were obtained by the turbo data depending scanning (TDDS) function. For quantitative purposes, calibration curves were obtained; rutin, chlorogenic acid, catechin and phloridzin were used for quantification of flavonoid, caffeoylquinic acid derivatives, proanthocyanidins and chalcone derivatives, respectively. Calibration curves were as follows: rutin y = 27,788x + 3307 (r² = 0.9981); chlorogenic acid y = 47,359x + 43,999 (r² = 0.9951); catechin y = 20, 525x + 32,962 (r² = 0.999) ; phloridzin y = 87,029x − 1,832 (r² = 0.999).

A previously published method was used for quantification of triterpene acids [34]. Briefly, an Agilent Eclipse XDB C-18 (3.0 × 150 mm, 3.5 μm) was used as stationary phase. Methanol (A) and H_2_O with 0.05 % formic acid (B) were the mobile phases. The analysis revealed the presence of annurcoic acid [34]; the latter was quantified using the calibration curve of the purified compound considering the ion species at *m/z* 485 in the range 5–50 µg/mL. The calibration curve was as follows, y = 12549x + 136925; r² = 0.9962.

### 4.4. Animals

This study was performed on male adult Wistar rats with a weight of 200–250 g (Tehran University of Medical Sciences, Iran). Animals were kept under standard conditions (12 h light/dark cycle; 23 °C ± 2) with free access to standard laboratory food and water. All procedures were done in accordance with the National Institute of Health Guide for the Care and Use of Laboratory Animals (NIH Publications No. 8023, revised 1978) [35]. All experiments were carried out at time between 9:00 to 14:00. Each group contained seven animals.

### 4.5. Chemicals

CCl_4_ was obtained from Merck; SOD and MPO Assay Kits were acquired from international and domestic commercial companies (ZellBio GmbH, Ulm, Germany). The serum levels of TNF-α and IL-10 were measured by the ELISA method using rat TNF-α ELISA Kit (ab46070, Abcam, Cambridge, UK) and rat IL-1β ELISA Kit (ab100768, Abcam, Cambridge, UK).

### 4.6. Experimental Design

Animals were divided in the following four groups (seven rats each):Control group: received only normal saline every dayCCl_4_ group: received CCl_4_ (25% CCl_4_-paraffin oil mixture, 2 mL/kg body weight, i.p (intraperitoneal); on the third day)APE group: received 30 mg/kg/day b.w (body weight). per os (oral administration) of APE and CCl_4_APP group: received 30 mg/kg/day b.w. per os of APP and CCl_4_

The APE and APP groups received daily the extract dose for three consecutive days. CCl_4_ was injected 3 h after the last treatment on the third day. Thereafter, 18 h later rats were anesthetized with ketamine (60 mg/kg ip) and xylazine (5 mg/kg ip) and sacrificed to collect heart, blood and liver for further analysis. 

### 4.7. Biochemical Analysis

Serum levels of ALT, AST, ALP, urea and creatinine were analyzed by using an autoanalyzer (COBAS Mira).

### 4.8. Evaluation of Antioxidant Markers

Antioxidant enzymes such as SOD, MPO and pro-inflammatory cytokines such as TNF-α and IL-1β were estimated from liver tissue homogenates of animals. Liver tissue was cut into some sections and homogenized in phosphate buffer (0.1 M, pH = 7.4). After centrifuging (4 °C at 4000 rpm, 20 min), the supernatant was collected to estimate the SOD and MPO levels using colorimetric enzyme Assay Kit (ZellBio GmbH); TNF-α and IL-1β were determined using colorimetric enzyme immunoassay (R&D Systems Inc., Biosource, Minneapolis, MN, USA).

### 4.9. Histopathological Examination

Liver sections were collected and fixed with neutral formalin 10%. Twenty-four h later, the samples were embedded in paraffin, and then sectioned with a microtome (RM2235 Rotary Microtome) to obtain 5 μm-thick paraffin sections. Thereafter, slices were stained with hematoxylin and eosin and the rate of tissue damages evaluated.

### 4.10. Statistical Analysis

The data are presented as mean ± SEM. Statistical analyses were done using SPSS software (version 21) (SPSS Inc., Chicago, IL, USA). One-way analysis of variance (ANOVA) followed by post hoc Tukey’s test were used to analyze the differences between groups. *p* Value < 0.05 was considered statistically significant for all analyses.

## 5. Conclusions

Based on our results, the hepatoprotective effects displayed by APE may be ascribed to the presence of a complex pattern of phytochemicals as bioactive components, such as quercetin derivatives, proanthocyanidins, dihydrochalcones and triterpene acids. In this respect, a recent study highlighted a higher content of these phytoconstituents in the Mela Rosa dei Monti Sibillini compared to other Italian apple varieties [36]. This may be of great importance to increase the possibility of using this old apple variety and its byproducts in the preparation of nutraceuticals helpful to manage liver disorders. At the same time, they give an added value to improve the regional economy through cultivation of orchards of this traditional apple.

## Figures and Tables

**Figure 1 molecules-25-01816-f001:**
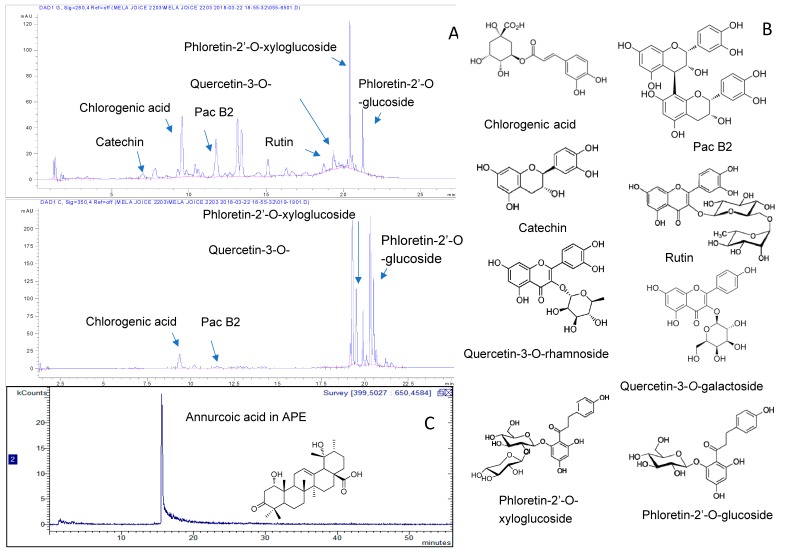
HPLC (High performance liquid chromatography) chromatogram of apple pulp (APP) at 254 nm (**A**), with chemical structures of the main constituents (**B**). LC-APCI-MS (liquid chromatography atmospheric pressure chemical ionization mass spectrometry) chromatogram of apple peel (APE) extract showing a peak related to annurcoic acid (**C**).

**Figure 2 molecules-25-01816-f002:**
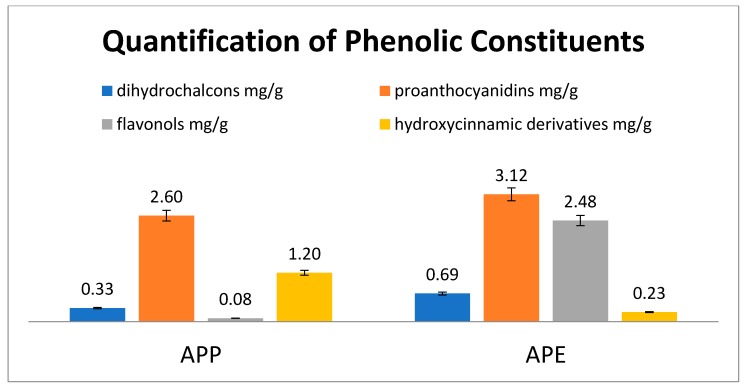
Amounts of different polyphenols in the investigated apple extracts.

**Figure 3 molecules-25-01816-f003:**
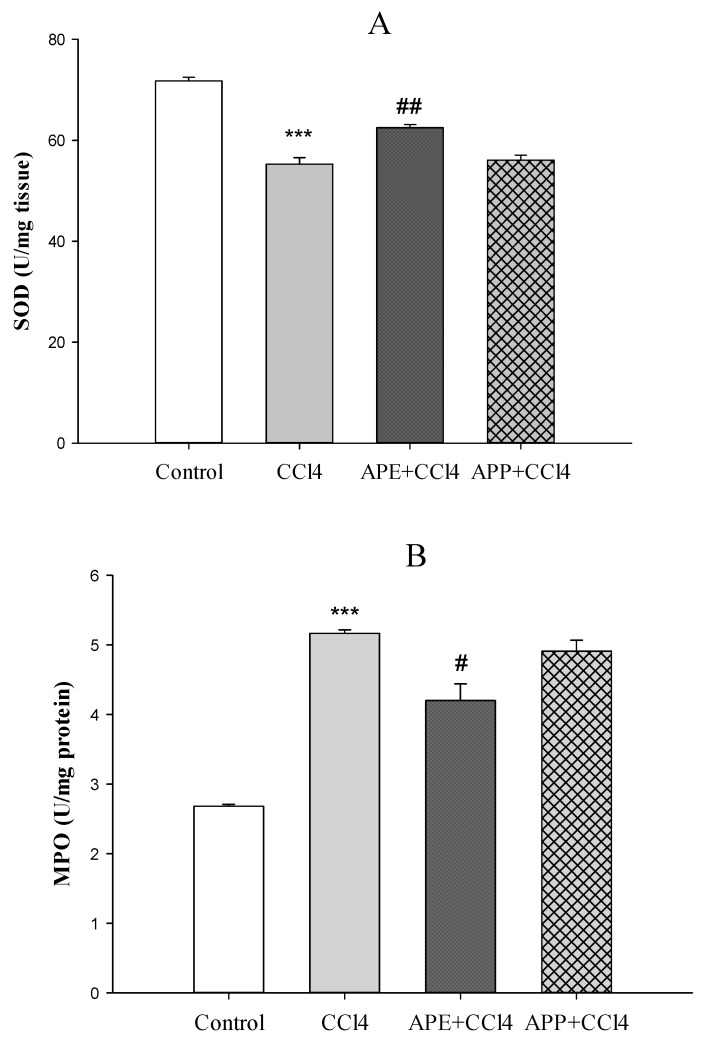
The effects of APE (30 mg/kg, oral), APP (30 mg/kg, oral), and CCl_4_ (2 mL/kg; ip (intraperitoneal administration) once on third day) administration on the SOD (**A**) and MPO (**B**) levels in liver tissue in rats. Data are expressed as mean ± SD. *** *p* < 0.001 compared to the control group, ^#^
*p* < 0.05, ^##^
*p* < 0.01 compared to the CCl_4_ group.

**Figure 4 molecules-25-01816-f004:**
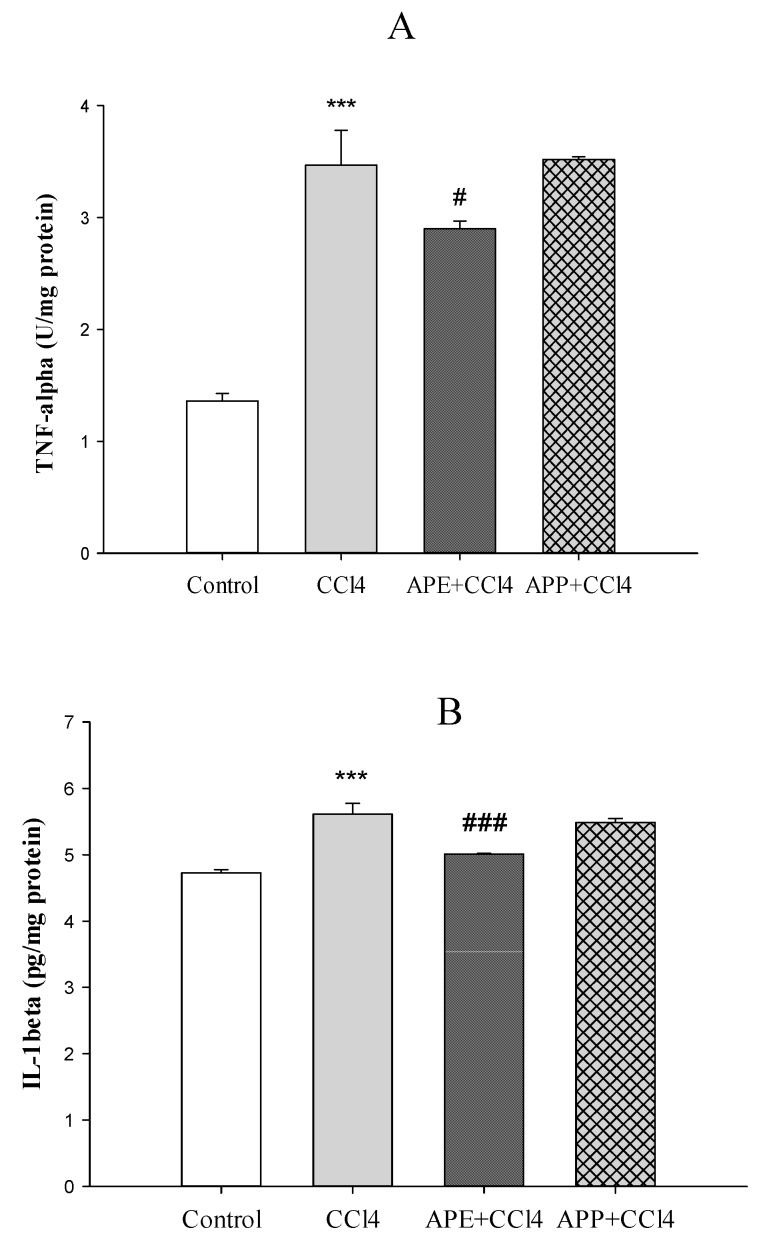
The effects of APE (30 mg/kg, oral), APP (30 mg/kg, oral), and CCl_4_ (2 mL/kg; ip once on third day) administration on the TNF-α (**A**) and IL-1β (**B**) levels in liver tissue in rats. Data are expressed as mean ± SD. ****p* < 0.001 compared to the control group, ^###^
*p* < 0.001, ^#^
*p* < 0.05 compared to the CCl_4_ group.

**Figure 5 molecules-25-01816-f005:**
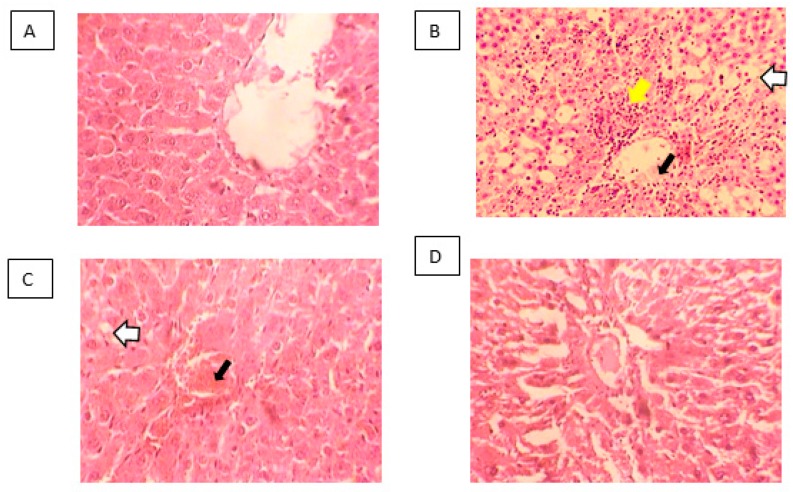
Micrographs of hematoxylin- and eosin-stained liver tissues are presented (×10). Control livers present normal histologic structures (**A**). CCl_4_ (**B**) and APP+CCl_4_ (**C**) treated livers show disrupted cell boundaries and cellular necrosis (white arrows), congestion of the central vein (black arrows) and inflammation (yellow arrow) whereas APE + CCl_4_ (**D**) livers show enhanced histopathologic features.

**Table 1 molecules-25-01816-t001:** Effects of APE and APP on serum biochemical parameters ^a^ in CCl_4_-induced hepatic damage in rats.

Examinations	Urea	Creatinine	AST	ALT	ALP
**Control**	50.28 ± 1.85	0.63 ± 0.07	125.5 ± 26.8	48 ± 6.46	316.1 ± 24.45
**CCl_4_**	53.10 ± 0.90	0.71 ± 0.10	633.0 ± 22.8 ^###^	632.2 ± 53.59 ^##^	599 ± 55.96 ^#^
**APE + CCl_4_**	46.00 ± 7.00	0.75 ± 0.03	230.5 ± 33.5 ^**^	181 ± 69 ^**^	361.5 ± 41.5 ^*^
**APP + CCl_4_**	37 ± 2.00	0.63 ± 0.07	151.5 ± 3.50 ^**^	110.5 ± 3.50 ^**^	375 ± 25 ^*^

^a^ Data are expressed as the mean ± SEM. ^#^
*p* < 0.05, ^##^
*p* < 0.01 and ^###^
*p* < 0.001 compared with control group. * *p* < 0.05 and ** *p* < 0.01 compared with CCl4 group.

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
