# Peer review of "Hepatoprotective Effects of Standardized Extracts from an Ancient Italian Apple Variety (Mela Rosa dei Monti Sibillini) against Carbon Tetrachloride (CCl4)-Induced Hepatotoxicity in Rats"

_molecules, 2020, doi:10.3390/molecules25081816_

Round 1
Reviewer 1 Report
The authors studied the hepatoprotective effects of standardized extracts (APP and APE) from an ancient Italian apple variety against carbon tetrachloride-induced hepatotoxicity in rats. They showed that the extracts ameliorated liver injury by reducing SOD activity and inflammation. The study is original, the methods are adequately described and the results are clearly presented. The conclusions are supported by the results.
The main weakness of the current study is a lack of analyses that could provide more accurate insight into the mechanism of the hepatoprotective activity of APP and APE.
I have a few suggestions that could improve the quality of the manuscript:
- The first part of the results in the manuscript is basically a repetition of previously published data. However, neither that or the current study presents data on the antioxidant activity of the extracts. Please determine the scavenging effects of the extracts on hydroxyl, superoxide and DPPH radicals, as well as their reducing power.
- What is the reason for measuring urea and creatinine? Please give a rationale.
- Please include relative liver weights in Table 1.
- It is surprising that AST and ALT levels were lower in the APP-CCl4 group than APE-CCl4 group, whereas the other parameters were lower in the APE-CCl4 group. Please comment this.
- In Fig. 4 symbols denoting a statistical difference are not clearly visible.
- The results in Fig. 5 are not presented properly. The CCl4 dosage (1:1 v/v in paraffin oil, 2 mL/kg) is very high. It should virtually erase hepatocytes around the central vein by a massive necrosis, which is not visible from the Fig. 5B. This figure also has a different magnification than the others. I suggest using figures with a lower magnification (x10) with the insets x40. Please mark the histopathological changes.
- The authors consider MPO as the antioxidant enzyme occasionally and as an index of neutrophil infiltration. Please clearly state the role of MPO. MPO is not the best choice for showing the antioxidant effects of the extracts. Please determine at least one additional marker of oxidative stress.
Author Response
Reviewer#1
The authors studied the hepatoprotective effects of standardized extracts (APP and APE) from an ancient Italian apple variety against carbon tetrachloride-induced hepatotoxicity in rats. They showed that the extracts ameliorated liver injury by reducing SOD activity and inflammation. The study is original, the methods are adequately described and the results are clearly presented. The conclusions are supported by the results.
The main weakness of the current study is a lack of analyses that could provide more accurate insight into the mechanism of the hepatoprotective activity of APP and APE.
I have a few suggestions that could improve the quality of the manuscript:
1. The first part of the results in the manuscript is basically a repetition of previously published data. However, neither that or the current study presents data on the antioxidant activity of the extracts. Please determine the scavenging effects of the extracts on hydroxyl, superoxide and DPPH radicals, as well as their reducing power.
OUR REPLY: this study represents a continuation of our studies on an ancient apple cultivar (Mela Rosa dei Monti Sibillini) cultivated in central Italy with the aim to improve its cultivation and valorize it for the development of nutraceuticals. Previously, we have demonstrated the antioxidant activities of its extracts both in vitro and in vivo (please see: Food & Function, 2019, 10.11: 7544-7552, Plants, 2020, 9.1: 9). That is the reason why in this manuscript we did not insert redundant data.
2. What is the reason for measuring urea and creatinine? Please give a rationale.
OUR REPLY: we inserted additional data to indicate which doses have no renal toxicity and only affects liver parameters.
3. Please include relative liver weights in Table 1.
OUR REPLY: we thank the reviewer for this comment. Weight of organ is commonly used for studies on lung tissues. For liver, histopathology and biochemical factors of liver function are normally examined.
4. It is surprising that AST and ALT levels were lower in the APP-CCl4 group than APE-CCl4 group, whereas the other parameters were lower in the APE-CCl4 group. Please comment this.
OUR REPLY: AST and ALT levels were lower in the APP-CCl4 group than APE-CCl4 group but there is not significant difference when the data were analyzed by SPSS. In Fig. 4 symbols denoting a statistical difference are inserted.
5. The results in Fig. 5 are not presented properly. The CCl4 dosage (1:1 v/v in paraffin oil, 2 mL/kg) is very high. It should virtually erase hepatocytes around the central vein by a massive necrosis, which is not visible from the Fig. 5B. This figure also has a different magnification than the others. I suggest using figures with a lower magnification (x10) with the insets x40. Please mark the histopathological changes.
OUR REPLY: we thank the reviewer for this comment. The dose of CCl4 which was used was 25% CCl4-paraffin oil mixture (2 mL/kg body weight). The figure of histopathology has been corrected and changed.
6. The authors consider MPO as the antioxidant enzyme occasionally and as an index of neutrophil infiltration. Please clearly state the role of MPO. MPO is not the best choice for showing the antioxidant effects of the extracts. Please determine at least one additional marker of oxidative stress.
OUR REPLY: A plethora of evidences showed that MPO is a more reliable factor in hepatoprotective studies and served as factor of inflammation; thus, we analyzed MPO activity to indicate the inflammatory response in liver, an index of neutrophil infiltration and also pro-inflammatory cytokines such as TNF-α and IL-1β. Besides, we analyzed SOD as an antioxidant enzyme.
Reviewer 2 Report
The manuscript Hepatoprotective effects of standardized extracts from an ancient Italian apple variety (Mela Rosa de Monti Sibillini) against carbon tetrachloride (CCL4)- induced hepatotoxicity in rats is an interesting document of the phenolic compounds presented in this important Italian apple variety in the peel and pulp, and its effect against carbon tetrachloride. The manuscript is well presented, described and discussed. Some minimum changes must be performed before its publication
Although the authors indicated that in previous work the detailed chemical analysis of the extracts was performed, in this document figure 1 could be presented in a different way to give a must detailed explanation of the compounds presented in both extracts with a better resolution of the chromatograms.
An in vitro study of the groups of compounds evaluated induvial could improve the discussion of the interesting results presented in the manuscript and confirmed the conclusion presented.
The symbols in figure 3 and 4 presented above the bars are not clearly defined in the captions.
Some minor mistakes as in H2O in line 273 need to be changed
Author Response
Reviewer#2
The manuscript Hepatoprotective effects of standardized extracts from an ancient Italian apple variety (Mela Rosa de Monti Sibillini) against carbon tetrachloride (CCL4)- induced hepatotoxicity in rats is an interesting document of the phenolic compounds presented in this important Italian apple variety in the peel and pulp, and its effect against carbon tetrachloride. The manuscript is well presented, described and discussed. Some minimum changes must be performed before its publication
1. Although the authors indicated that in previous work the detailed chemical analysis of the extracts was performed, in this document figure 1 could be presented in a different way to give a must detailed explanation of the compounds presented in both extracts with a better resolution of the chromatograms.
OUR REPLY: we changed the Fig. 1 and indicated the main peaks in the chromatograms of extracts.
2. An in vitro study of the groups of compounds evaluated induvial could improve the discussion of the interesting results presented in the manuscript and confirmed the conclusion presented.
OUR REPLY: We thank the reviewer for this comment. Due to COVID-19 outbreak, we have limited access to our lab. In the future, the in vitro study will definitely be done. Thanks again.
3. The symbols in figure 3 and 4 presented above the bars are not clearly defined in the captions.
OUR REPLY: we corrected the definition of bars in the captions of figure 3 and 4.
4. Some minor mistakes as in H2O in line 273 need to be changed
OUR REPLY: we made the suggested correction.
Round 2
Reviewer 1 Report
The authors made some improvements to the manuscript, but many questions still remain to be answered.
- What is the reason for measuring urea and creatinine? Please give a rationale in the manuscript. I could not see additional data indicating which doses have no renal toxicity and only affects liver parameters.
- Please include relative liver weights in Table 1.
- The results in Fig. 5 are not presented properly. There are still questions about the magnification of figures. Please insert a scale bar into the figures. Please mark all histopathological changes (necrosis, inflammation, etc.). Regarding fibrosis, it seems that figure 5B shows a vascular connective tissue (tunica adtventitia). Fibrosis can not be reliably determined from HE staining. Mallory's or Masson's trichrome stain should be used for detection of fibrous tissue. Alternatively, you can exlude fibrosis from the results and discussion, if you have problems with using lab due to the corona situation.
- Please state the function of MPO, emphasizing its dual role, as both antioxidant and anti-inflammatory enzyme.
Author Responsec
1. What is the reason for measuring urea and creatinine? Please give a rationale in the manuscript. I could not see additional data indicating which doses have no renal toxicity and only affects liver parameters.
In introduction we added the following sentence as a rationale.
During CCl4 intoxication, serum creatinine and urea levels is often increased by impairment of glomerular filtration rate due to the delayed CCl4 elimination [1]. Therefore, urea and creatinine levels were also measured in order to assess nephroprotective effects of our treatments. Fahmy, M.A., et al., Carbon tetrachloride induced hepato/renal toxicity in experimental mice: antioxidant potential of Egyptian Salvia officinalis L essential oil. Environmental Science and Pollution Research, 2018. 25(28): p. 27858-27876.
2.Please include relative liver weights in Table 1.
Unfortunately, we didn’t measure this parameter according to the standard procedures followed in most of the previously published articles. Furthermore, due to the impossibility to access department in this period, we cannot perform this measurement with a new group of animals.
3. The results in Fig. 5 are not presented properly. There are still questions about the magnification of figures. Please insert a scale bar into the figures. Please mark all histopathological changes (necrosis, inflammation, etc.). Regarding fibrosis, it seems that figure 5B shows a vascular connective tissue (tunica adtventitia). Fibrosis can not be reliably determined from HE staining. Mallory's or Masson's trichrome stain should be used for detection of fibrous tissue. Alternatively, you can exlude fibrosis from the results and discussion, if you have problems with using lab due to the corona situation.
The micrograph on CCL4 liver has been changed and corrections were done (see below). We excluded the fibrosis from our manuscript, due to impossibility to access laboratory (Covid-19 epidemic) and check it.
4. Please state the function of MPO, emphasizing its dual role, as both antioxidant and anti-inflammatory enzyme.
We inserted a reference relative to the key role of MPO during hepatic damage.
De Andrade, K. Q., Moura, F. A., Dos Santos, J. M., De Araújo, O. R. P., de Farias Santos, J. C., & Goulart, M. O. F. (2015). Oxidative stress and inflammation in hepatic diseases: therapeutic possibilities of N-acetylcysteine. International journal of molecular sciences, 16(12), 30269-30308.